# Women’s Perception of Male Involvement in Antenatal, Childbirth and Postnatal Care in Urban Slum Areas in Bangladesh: A Community-Based Cross-Sectional Study

**DOI:** 10.3390/healthcare9040473

**Published:** 2021-04-16

**Authors:** Muhammad Zakaria, A. K. M. Ziaur Rahman Khan, Md. Sarwar Ahmad, Feng Cheng, Junfang Xu

**Affiliations:** 1Department of Communication and Journalism, University of Chittagong, Chittagong 4331, Bangladesh; zakaria@cu.ac.bd (M.Z.); ziakhan@cu.ac.bd (A.K.M.Z.R.K.); 2Department of Mass Communication and Journalism, Begum Rokeya University, Rangpur, Rangpur 5404, Bangladesh; sarwarmcj@brur.ac.bd; 3Vanke School of Public Health, Tsinghua University, Beijing 100084, China; fcheng@mail.tsinghua.edu.cn; 4Center for Health Policy Studies, School of Public Health, Zhejiang University School of Medicine, Hangzhou 310058, China

**Keywords:** male involvement, maternal health, antenatal care, delivery care, postnatal care, slum areas

## Abstract

Male participation in reproductive health issues has been considered to be an effective and promising strategy to address the women’s reproductive health problems since the 1990s. Under this background, we aim to explore the women’s perception of men’s involvement in antenatal care (ANC), delivery and postnatal care (PNC) in the slum community of Bangladesh where various sexual and reproductive health problems exist. A community-based cross-sectional study was conducted among women and their husbands living in 12 slums of Chattogram city. Cross-tabulation with chi-square tests and multivariate logistic regression analyses were performed to examine the predictors of husbands’ support in wives’ antenatal, delivery and postnatal care. The study demonstrates that the education and economic level of most women and their husbands were very low although husbands seemed to have a better status than wives in these aspects. Almost all men (~90%) had never accessed services related to reproductive and maternal health. Only 10% of respondents gave birth to their last baby in government hospitals or private clinics. In addition, 60% of the husbands took care of their wives during pregnancy with 44% during childbirth and about 30% providing help in receiving postpartum care. Moreover, husbands’ discussions with a health worker regarding maternal and reproductive health were the most important predictors for support of their wives during pregnancy, childbirth and postpartum care (*p* < 0.05). Study participants’ perception of a satisfying spousal relationship also appeared to be a significant factor for husbands’ responsible role regarding wives’ antenatal care, delivery and postnatal care (*p* < 0.05). This study found that pregnant women living in slums received poorer health-related services when there was a low involvement of men; specifically, the husbands of pregnant women. In addition, men’s involvement was influenced by many aspects, particularly awareness-related factors (e.g., knowledge, communication and access to reproductive health services). Therefore, awareness creation is important for active involvement in antenatal, delivery and postnatal care. Strategies should be designed to provide men living in the slums with adequate information, education and communication to gain their interest and support about reproductive and maternal health.

## 1. Introduction

In recent years, global urbanization has become a major concern. As with many other developing countries, Bangladesh is facing various challenges due to urbanization in the last few decades [1]. About 67% of the urban population growth of Bangladesh is caused by the influx of rural residents migrating to cities for economic opportunities [2]. According to the Census of Slum Areas and Floating Population 2014, in Bangladesh, 22.32 million people lived in slum areas, which accounted for 6.33% of the urban population and 1.48% of the total population of the country. Chattogram city is home to 21.44% of households living in slum areas of the country [3]. 

Together with the growth of urban slums, this rapid urbanization in Bangladesh is likely to have profound consequences for its health profile, particularly for maternal and child health [4]. For example, many women die in the slums during pregnancy and childbirth. In addition, the mortality of children younger than five years in slums is almost double that of rural areas. Two-thirds of these deaths could be prevented if proper care was made available including regarding men’s participation in antenatal, delivery and postnatal care. Indeed, focusing on men’s motivations and responsibility in reproductive health received great momentum following the Program of Action formed at the 1994 International Conference on Population and Development (ICPD) [5]. The participation of men has been proven to be an important driving factor for the betterment of women’s reproductive health. In fact, in many countries, men often play a vital and dominant role in decision-making regarding family planning and reproductive health, which can lead to a significant impact on women’s health [6]. For example, men can facilitate the prevention of unintended pregnancies, promote safe motherhood, perform responsible fatherhood, not abuse women and reduce the spread of HIV and other sexually transmitted diseases [7]. 

As a country of a patriarchy, Bangladesh still faces many challenges in engaging male partners in family planning as well as reproductive health services despite having adopted the approach of involving men in reproductive health and maternal health in line with the agenda of the International Conference on Population and Development (ICPD) in Cairo in 1994 [8]. However, men are the key decision-makers in most sexual and reproductive health-related affairs in the household [9]. For example, about one-third (30.6%) of women said that their husbands alone made the decisions relating to their reproductive health care [10]. In most cases, men have considerable influence over women’s time and mobility in Bangladesh. Therefore, the place of delivery (e.g., at home or institutional) was mostly decided by men. However, there is no empirical evidence on the extent of male involvement in maternal and reproductive health in low resource settings in Bangladesh, which may limit the ability of health policy makers to design programs and initiatives to improve women’s health. Therefore, this study aimed to explore the women’s perceptions about their husbands’ involvement in antenatal care, childbirth and postnatal care and associated factors in urban slums of Chattogram city in Bangladesh.

## 2. Materials and Methods

### 2.1. Study Design, Setting and Participants

A community-based cross-sectional survey was used to collect the data from married women living in slums of Chattogram city, the commercial capital and the second largest city of Bangladesh with a population of more than 3.9 million [11]. The slum areas were selected based on the type of slums (i.e., ‘high’ and ‘low’ performing slum areas in terms of availability of utility services and infrastructure) to ensure participants from various socio-economic backgrounds and amenities would be included in the study. Finally, data were collected from six low performing slums and six high performing slums from 2216 slums of Chattogram city [3]. 

Married women who met the following criteria were incorporated into our study as the main respondents: those of reproductive age (15–49 years old), living in the slums of Chattogram city during the time of the survey, living with a husband and had given birth to at least one child at least one year preceding the time of the survey. All parents were married and women and men typically have children with only one partner in Bangladesh’s perspective. Therefore, man and woman indicate husband and wife, respectively, throughout the study. However, in this study, data about men’s participation in reproductive health was collected from their wives as the men who did not play a responsible role in maternal health may not provide accurate information, which would make the findings flawed and reduce the accuracy of the data. Finally, 200 participants were each investigated from high and low performing slums (N = 400). A total of 422 potential participants were approached, indicating a 94.79% response rate. Study participants were drawn from each slum using convenience sampling due to an insufficient budget. Before giving out the questionnaire to the study participants, the data collectors explained to them the importance and the goal of this study. The sample size was determined using a single population proportion formula considering the following assumptions: *p* = 50% expected a satisfactory level of husbands’ participation in wives’ maternal health, significance level 5% (α = 0.05), Z $\frac{\alpha }{2}$ = 1.96, margin of error 5% (d = 0.05) and a 10% non-response rate.

### 2.2. Data Collection

Data were collected using a pretested, structured and facilitator-administered questionnaire. The questionnaire consisted of four parts: (a) women’s socio-demographic characteristics; (b) husbands’ socio-demographic characteristics; (c) communication, knowledge and access to health service-related variables; (d) women’s perception of husbands’ support in their antenatal care (ANC), delivery and postnatal care (PNC). Postnatal care implied receiving care from a medically trained provider just after the delivery or within two days of delivery and up to 42 days after the birth of the baby. The questions related to the women’s perception of male involvement in maternal health were finalized after a pilot study was carried out with 30 participants to ensure the quality of the questionnaire. 

### 2.3. Measure of Outcome Variables

Regarding the women’s perception of male participation in antenatal care, a score was assigned to each question. Each positive answer was assigned the score ‘1′ or ‘0′ if negative. The total score of all questions was then calculated and dichotomized using the mean as a cut-off value. A score below the mean value was then coded as ‘0′ reporting a low involvement and above the mean value was coded as 1 indicating a high involvement with regard to antenatal care [12,13,14]. Likewise, wives’ perception of husbands’ involvement regarding wives’ delivery and postnatal care were measured similarly with a low and a high involvement. 

### 2.4. Statistical Analysis

Descriptive statistics were used to describe husbands’ support in wives’ antenatal care, delivery and postnatal care and their socio-demographic, communication, knowledge and access to health service-related characteristics. Variables with a *p* < 0.05 in the chi-square test were included in the multivariate logistic regression model to examine the predictors of husbands’ support in wives’ antenatal, delivery and postnatal care. The Omnibus Tests of Model Coefficients gives an overall indication of the ‘goodness of fit test’ and reported that our multivariate regression model performed well and would be a good predictor of the three outcome variables. Model fits for predicting husbands’ support regarding wives’ antenatal care were χ2 (16) = 128.46 (*p* < 0.001) and Nagelkerke *R^2^* = 0.37; husbands’ assistance as to wives’ delivery were χ2 (15) = 150.75 (*p* < 0.001) and Nagelkerke *R^2^* = 0.42; husbands’ support relating to wives’ postnatal care were χ2 (16) = 165.69 (*p* < 0.001) and Nagelkerke *R^2^* = 0.47. Variables with a *p*-value < 0.05 were taken as the significant factors. The statistical software SPSS 24.0 software (IBM, Armonk, NY, USA) was used to analyze all data.

## 3. Results

### 3.1. Socio-Demographic Characteristics of the Respondents and Their Husbands

Table 1 shows the socio-demographic characteristics of respondents and their husbands. The mean of education years of the respondents was 2.79. Of them, 202 (50.5%) had no formal education and 238 (59.5%) were housewives. The mean education year of respondents’ husbands was 4.06 and 151 (37.8%) had no formal education; 178 (44.6%) were involved in small businesses. Among the study participants, 131 (32.8%) had a monthly household income of greater than BDT 5000–10,000. The mean age of respondents was 25.51 (SD = 5.72) while the mean age of their husbands was 32.24 years (SD = 6.63). Regarding the number of children of respondents, 146 (36.5%) had one child followed by 144 (36%) who had two children and 110 (27.5%) who had more than two children.

### 3.2. Respondents’ and Their Husbands’ Access to Maternal and Reproductive Health Services

Table 2 demonstrates the different variables relating to respondents’ and their husbands’ access to maternal and reproductive health services and programs. More than one-third (36.8%) reported that the government of Bangladesh (GoB) and NGOs ever implemented family planning and maternal health awareness programs in the slums where they were living. Furthermore, 4.3% of the respondents’ husbands participated in any program on family planning and maternal health issues and less than one-third of respondents’ husbands (31.3%) had a discussion with NGO/health workers about maternal and reproductive health.

### 3.3. Delivery Location and Birth Attendants during the Last Delivery of Respondents

Figure 1 shows the delivery locations and birth attendants during the respondents’ most recent childbirth. More than half (57%) of respondents’ last delivery had taken place at slum houses. In contrast, only 10% of respondents gave birth to their last baby in government hospitals or private clinics. In addition, only 27% of respondents’ deliveries were assisted by a licensed doctor whereas more than one-third (36%) of deliveries were assisted by a midwife.

### 3.4. Male Participation in Antenatal, Delivery and Postnatal Care

Figure 2 demonstrates that 59.8% of respondents perceived that husbands provided a high level support in wives’ antenatal care. Furthermore, 44% of respondents had a perception that husbands provided a high level of assistance in their wives’ delivery care while 35.8% provided a high level of participation in postnatal care.

Table 3 depicts the women’s perception of different types of support provided by husbands during the antenatal, delivery and postnatal periods. It showed that although about 70% of husbands took care of wives’ nutrition during ANC (78.7%) and PNC (67.5%) only one third of husbands managed to provide the total expense to visit a doctor (34.7% for ANC and 35.3% for delivery). Moreover, only 35% of husbands took their wives to the health center or a hospital for delivery.

### 3.5. Predictors of the Husbands’ Participation in Wives’ Antenatal, Delivery and Postnatal Care

Table 4 reveals that variables related to socio-economic, communication and knowledge had been found to be statistically significant predictors for the women’s perception of their husbands’ high support in maternal health.

Husbands’ employed as drivers, service workers or garment workers (OR = 2.07, 95% CI: 1.16–3.68), women’s perception of having a good matrimonial relationship (OR = 2.53, 95% CI: 1.50–4.26), having a moderate knowledge on RH and MH (95% CI: 1.12–4.26), living in a slum where NGOs had RH- and MH-related programs (OR = 1.89, 95% CI: 1.00–3.55) and husbands who had previously discussed maternal and reproductive health issues with a health worker (OR = 7.00, 95% CI: 3.33–14.70) demonstrated a higher likelihood of providing antenatal care than their counterparts (*p* < 0.05).

Regarding men’s assistance in delivery care, respondents who had more than two children were 35% (95% CI: 0.16–0.75) less likely to have their husbands’ help in delivery care than those who had one child (*p* < 0.05). In addition, couples that had greater utility facilities (gas, water and power supply) in the slum areas, women’s perception of having a good marital relationship (OR = 2.18, 95% CI: 1.27–3.75), couples who discussed reproductive health issues together and husbands who had previously discussed maternal and reproductive health issues with a health worker had a more significant association with husbands’ positive role during delivery (*p* < 0.05).

The statistically significant predictors on the help provided for postnatal care also showed similar results with delivery care. In addition, the likelihood of husbands helping with wives’ postnatal care among husbands who watched TV was 2.08 times higher relative to husbands who did not watch TV (95% CI: 1.20–3.60) and an increasing number of children saw a lower likelihood for the husbands’ responsible role in wives’ PNC (OR = 0.36, 95% CI: 0.18–0.69) with *p* < 0.05.

## 4. Discussion

In our study it was noticeable that a cohort of husbands were not concerned about wives’ maternal health-related medical check-ups especially with regard to providing company and managing the expenses. Husbands’ support in wives’ maternal health was lower than those of non-slum areas [15]. This might be due to a lack of awareness, which was demonstrated in our findings.

Our study revealed that only 10% of births took place in hospitals or clinics while 57% took place in the home. In addition, 33% of respondents reported that they delivered babies in the houses of nurses who worked in hospitals or clinics because those nurses who assisted in childbirth charged a lower cost than hospitals or clinics, apart from the issue of legality. It was found that only 41% of deliveries were assisted by doctors or nurses, which was in line with the data of BDHS [16]. In fact, in Bangladesh, 30% of women cannot go to the hospital or health center alone or accompanied by their children because of their husbands’ disapproval [16]. It was also found that husbands were reluctant not only to make antenatal care available but also to support wives’ delivery in a health facility and to hire a skilled birth attendant (SBA). Only a small portion of husbands (35.8%) provided a high level of support in maternal health. This finding was lower than others [17,18,19,20]. For example, in a study in rural Bangladesh, Rahman et al. [17] found that 47% of women who attended an ANC visit were accompanied by their husbands and approximately half of the husbands were present at the time of childbirth while 67% were with their wives during a PNC visit. Similarly, in a low resource setting in Tanzania, Gibore et al. [18] reported that male involvement was high in terms of accompanying partners to ANC, providing physical support during pregnancy and making joint decisions for ANC. In a study among married men in Nigeria, Falade-Fatila and Adebayo [19] observed different levels of involvement in various domains of pregnancy-related care with the highest levels of involvement recorded in the areas of reminding and financial support while their participation in conducting treatment tasks such as accompanying their partner to clinic visits was very low. Furthermore, Adams et al. [20] found that about half of farmers who lived in rural communities in central Malawi went with their wives for their one week PNC visits.

In our study, the number of children appeared to be a predictor of male participation in maternal health. The results depicted the inverse relationship between the number of children and men’s active role. Respondents who had more than two children experienced a lower level of participation by their husbands in reproductive matters. This related to the fact that nowadays low-income families living in urban areas want fewer children due to their perception that having more children might hinder economic prosperity and wealth. Similarly, slum inhabitants are also able to comprehend small family norms implying that having a larger family would increase their level of poverty amid the government integrated population and family planning program that has been implemented over the last few decades. This is why husbands with larger families may become less interested in supporting their wives during pregnancy and delivery. Nevertheless, the consequence of this attitude may lead to adverse health outcomes for their wives and neonatal health as the husband’s support is essential to promote safe motherhood. Our findings contradicted a previous study [21] that found a positive association between male participation and the number of children born.

Access to media is one of the determining factors of being aware about the importance of participation in maternal health. This study reported a significant relationship between male involvement and the amount of time they watched TV. Since the invention of television, it has been a powerful, intrusive, attractive and ubiquitous medium [22]. Television, being both a “news and entertainment medium” [23], disseminates information, increases knowledge, influences attitudes, beliefs and behaviors and transmits values to its viewers [24]. Our findings were similar to previous studies conducted in Bangladesh [8,21,25]. Mass media often disseminates information and content on reproductive and maternal health issues from which audiences can obtain tailored information and necessary knowledge. Hence, a positive association was found between husbands’ habits of television watching (*χ2* = 35.89, *p* < 0.001) and their knowledge on maternal health.

As was observed, men living in slums with more utility services and high performing areas were more supportive regarding their wives’ pregnancy and post-delivery care than those who lived in low performing areas. Our study depicted that relationships were an important factor for the level of husbands’ participation in wives’ maternal health. Usually, couples who have a good relationship are open to each other and can share their feelings, anxieties, necessities and opinions. Couples that maintain a good marital relationship are more likely to support each other as they become more spontaneous and dedicated to helping their partner. Similarly, through inter-spousal communication, husbands and wives can share information, ideas and problems about reproductive health, resulting in enhancing the status of women’s health. Accordingly, this is considered to be an essential catalyst of male motivation toward reproductive health participation and awareness [7]. As observed in our study, couple communication on reproductive health appeared to be a highly significant factor associated with men’s support for family planning and maternal health. This finding was in line with that of various other studies in developing countries [21,26].

Knowledge is an inevitable prerequisite for the forming of a favorable attitude and practicing recommended behavior in the health sector [27]. This is why increasing knowledge is one of the primary goals of all health communication interventions [28,29]. A learned person is better informed about his/her duties; for instance, wives who are knowledgeable about pregnancy and delivery-related complications expect to receive support from their husbands in family planning and antenatal care. This study found that respondents’ knowledge was a significant predictor of husbands’ participation regarding maternal health care. Our finding was consistent with a previous study [19].

The presence of Government of Bangladesh (GoB) and NGO awareness programs on reproductive and maternal health in slum areas appeared to be a factor influencing husbands’ support of wives during pregnancy. We found that the awareness programs by government and non-government organizations (NGOs) on reproductive health and safe motherhood and access to health services were inadequate. Only about one-third of slum dwellers were found to have come into contact with NGO awareness programs and visited hospitals or health centers for wives’ reproductive and maternal health care. In addition, less than one-third of husbands had communication with health workers and participated in programs on family planning and reproductive and sexual health. A lack of awareness of programs and the unavailability of health services can result in poor knowledge and low maternal health status. This explanation was supported by the findings of this study that illustrated a significant association (*p* < 0.001) between explanatory and outcome variables. A husband can be informed of different essential issues of women’s maternal and reproductive health and be motivated to offer support to their wives through interactions with health care providers. Generally in Bangladesh, while a pregnant woman visits the health center or clinic for an antenatal check-up, the physician or other health worker advises the male partner accompanying the pregnant woman on how to support his wife so that pregnancy-related complications can be avoided and wives’ delivery would take place in a setting with adequate facilities in the presence of a skilled health care provider. However, our finding was similar with another study [30].

## 5. Conclusions

We found that pregnant women living in slums received worse-related health services when there was a low involvement of men; specifically, the husbands of pregnant women. In addition, men’s involvement was influenced by many aspects especially the related awareness factors (e.g., knowledge, communication and access to reproductive health services). Considering the growing numbers of urban poor living in slum settlements, action is needed to tackle the adverse social determinants of women’s health and increase access to maternal healthcare services. For example, the GoB and NGOs should strengthen MH and SRH-related awareness programs in the slum areas, communicating with male audiences through relevant and attractive concepts and messages regarding the importance of their involvement in maternal and reproductive health. Information, education and communication with regard to reproductive and maternity care can be useful to gain men’s interest, support and approval about reproductive and maternal health. Such programs can reach more men in the places where men gather together regularly such as the workplace, clubs, parks and fields and men may be more attentive and receptive to new messages in these places.

### Limitations

This study had several limitations. First, there are a vast number of slums across Bangladesh with varying levels of overall economic conditions and health services. Only 12 slums located in Chattogram city were selected, which might impact the generality and applicability of our results. Second, in our study, male involvement in the female’s family planning and maternal health were depicted based on wives’ perceptions, which might be subject to reporting errors. Third, the findings could also be affected by recall bias.

## Figures and Tables

**Figure 1 healthcare-09-00473-f001:**
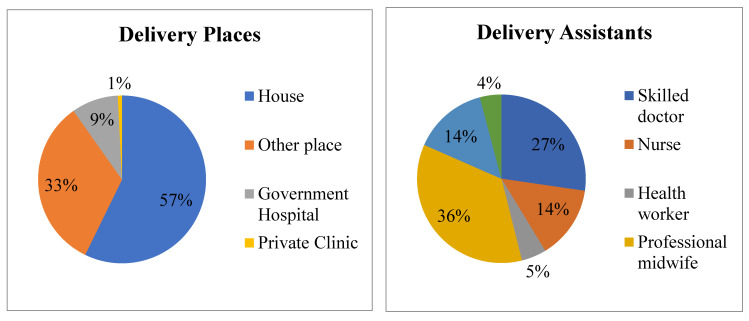
Distribution of the delivery places (**left**) and assistants of the respondents’ last childbirth (**right**).

**Figure 2 healthcare-09-00473-f002:**
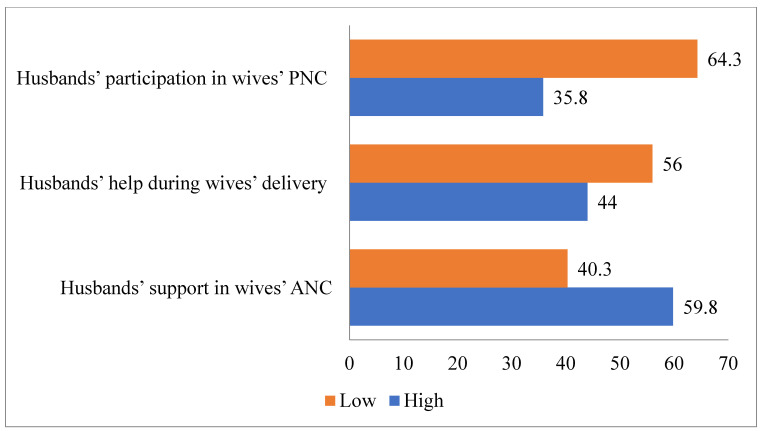
Distribution of women’s perception of husbands’ involvement level in maternal health care.

**Table 1 healthcare-09-00473-t001:** Distribution of the socio-demographic characteristics of respondents and their husbands.

Variables	Frequency	Percentage
Respondents’ education (mean years 2.79, SD = 3.30)		
No education	202	50.5
Up to class 5	117	29.3
>class 5	81	20.3
Husbands’ education (mean years 4.06, SD = 2.77)		
No education	151	37.8
Up to class 5	113	28.3
>class 5	136	34.0
Respondents’ occupation		
Housewife	238	59.5
Day laborer	13	3.3
Business/domestic worker	90	22.6
Service/garment worker	59	14.8
Husbands’ occupation		
Day laborer/rickshaw puller/jobless	116	26.6
Business	178	44.6
Service/garment worker	82	20.5
Driver	34	8.5
Respondents’ age (mean ± SD)	25.51 ± 5.72	
Husbands’ age (mean ± SD)	32.24 ± 6.63	
Monthly household income		
≤5000 BDT (≤60 USD)	59	14.8
5000–10,000 BDT (60–118 USD)	131	32.8
10,001–15,000 BDT (119–176 USD)	129	32.3
≥15,000 BDT (≥176 USD)	81	20.3
Number of children		
1	146	36.5
2	144	36.0
>2	110	27.5
Respondents’ TV viewing		
Yes	261	65.3
No	139	34.8
Husbands’ TV viewing		
Yes	201	50.3
No	199	49.8
Respondents’ radio listening		
Yes	61	15.3
No	338	84.5
Husbands’ radio listening		
Yes	101	25.3
No	299	74.8
Respondents’ internet use		
Yes	34	8.5
No	364	91.0
Husbands’ internet use		
Yes	130	32.5
No	270	67.5
Having a utility facility in the slum		
Little	200	50.0
More	200	50.0

Note: SD: standard deviation; BDT: Bangladeshi Taka; USD: US Dollar.

**Table 2 healthcare-09-00473-t002:** Distribution of respondents’ and their husbands’ access to FP, RH and MH services.

Variables	Yes (*n* (%))	No (*n* (%))
Ever had NGOs RH and MH awareness in slum areas	147 (36.7)	253 (63.3)
Husbands ever participated at any meeting on RH and MH	17 (4.3)	383 (95.7)
NGOs ever supplied contraceptives in slum areas	8 (2.0)	392 (98.0)
Husbands ever discussed with NGO/health worker on FP and RH	125 (31.3)	275 (68.7)

Note: FP: family planning; MH: maternal health; RH: reproductive health.

**Table 3 healthcare-09-00473-t003:** Distribution of wives’ perception of husbands’ support in different aspects of ANC, delivery and PNC.

Items	Yes (*n* (%))	No (*n* (%))
Took care of wife’s nutrition and rest during pregnancy	315 (78.7)	85 (21.3)
Reminded the wife about ANC visits	21 (5.3)	379 (94.2)
Managed the total expense to visit the doctor for ANC check-up	139 (34.7)	261 (65.3)
Accompanied wife during visiting doctor for ANC check-up	228 (57.0)	172 (43.0)
Advised and took initiative for going to hospital/clinic during delivery	19 (4.8)	381 (95.2)
Managed the total expense to go to the health center during delivery	141 (35.3)	259 (64.8)
Took wife to the health center or hospital for delivery	140 (35.0)	260 (65.0)
Supported in arranging an SBA during delivery	211 (52.7)	189 (47.3)
Helped in visiting the doctor during the postnatal period	161 (40.3)	239 (59.7)
Took care of wife’s nutrition and rest during PNC	270 (67.5)	130 (32.5)

Note: ANC: antenatal care; PNC: postnatal care; SBA: skilled birth attendant.

**Table 4 healthcare-09-00473-t004:** Multivariate logistic regression predicting husbands’ involvement in wives’ ANC, childbirth and PNC.

Characteristics (*N* = 400)	ANC	Delivery	PNC
	OR (95% CI)	OR (95% CI)	OR (95% CI)
^⸸^ Husbands’ education			
No education (ref.)	1		
Have education	1.40 (0.83–2.35)		
Husbands’ occupation			
Labor/rickshaw puller/business (ref.)	1	1	1
Driver/service/garment worker	2.07 (1.16–3.68) ^a^	1.61 (0.91–2.84)	1.73 (0.95–3.15)
^⸸^ Husbands’ TV viewing			
No (ref.)	1		1
Yes	1.18 (0.69–2.00)		2.08 (1.20–3.60) ^b^
Number of children			
1 (ref.)	1	1	1
2	0.56 (0.30–1.04)	0.72 (0.39–1.31)	0.36 (0.18–0.69) ^b^
>2	0.70 (0.33–1.52)	0.35 (0.16–0.75) ^b^	0.70 (0.31–1.58)
^⸸^ Having a utility facility in the slum			
Little (ref.)		1	1
More		3.10 (1.61–5.96) ^c^	2.33 (1.17–4.64) ^a^
Relationship between husband and wife			
Not good (ref.)	1	1	1
Good	2.53 (1.50–4.26) ^c^	2.18 (1.27–3.75) ^b^	4.61 (2.43–8.76) ^c^
Couple communication on FP and RH			
Rare (ref.)	1	1	1
Sometimes	1.75 (0.97–3.14)	1.96 (1.04–3.73) ^a^	1.46 (0.73–2.91)
Often/regular	1.48 (0.74–2.98)	2.81 (1.34–5.91) ^b^	2.20 (0.98–4.94)
Starting time of couple communication on FP and RH			
After marriage (ref.)	1	1	1
After the birth of the first baby	1.33 (0.72–2.47)	1.41 (0.77–2.57)	1.28 (0.67–2.45)
After the birth of the second baby	1.33 (0.62–2.86)	0.82 (0.37–1.84)	0.55 (0.22–1.39)
Respondents’ perception of husbands’ FP, RH and MH knowledge			
Poor (ref.)	1	1	1
Moderate	0.62 (0.33–1.15)	0.70 (0.36–1.36)	0.48 (0.23–1.02)
Good	0.48 (0.16–1.44)	1.18 (0.41–3.36)	0.43 (0.14–1.32)
Respondents’ self-reported FP, RH and MH knowledge			
Poor (ref.)	1	1	1
Moderate	2.19 (1.12–4.26) ^a^	1.40 (0.66–2.99)	1.03 (0.45–2.39)
Good	1.90 (0.57–6.36)	1.29 (0.38–4.37)	2.01 (0.57–7.08)
NGO’s RH and MH program in the slum			
No (ref.)	1	1	1
Yes	1.89 (1.00–3.55) ^a^	0.75 (0.40–1.41)	0.87 (0.45–1.69)
Husbands ever discussed with a health worker about RH and MH issues			
No (ref.)	1	1	1
Yes	7.00 (3.33–14.70) ^c^	5.55 (2.89–10.66) ^c^	5.52 (2.92–10.43) ^c^
Model chi-square	128.46 ^c^	150.75 ^c^	165.69 ^c^
−2 log likelihood	410.75	397.99	355.88
Nagelkerke *R^2^*	0.37	0.42	0.47

Note: ^⸸^ The variables were not included for adjusted logistic regression predicting the outcome variables as *p*-values were > 0.05 in bivariate analyses. OR = odds ratio; CI = confidence interval; ref. = reference category; ANC = antenatal care; PNC = postnatal care; FP = family planning; RH: reproductive health; MH: maternal health. ^a^
*p* < 0.05, ^b^
*p* < 0.01, ^c^
*p* < 0.001.

## Data Availability

All of the primary data were included in the results. Additional materials with details may be obtained from the corresponding author.

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
