# Peer review of "Women’s Perception of Male Involvement in Antenatal, Childbirth and Postnatal Care in Urban Slum Areas in Bangladesh: A Community-Based Cross-Sectional Study"

_healthcare, 2021, doi:10.3390/healthcare9040473_

Round 1

Reviewer 1 Report

need to reference similar populations in other countries, emphasis on successful programs, otherwise ok to publish

Author Response

Response to the Comments of Reviewer #1:

  1. Need to reference similar populations in other countries, emphasis on successful programs, otherwise ok to publish

Response: Thank the reviewer’s helpful suggestion. We have now carefully revised the discussion section by adding some literature in other countries. Please see page 8.

Reviewer 2 Report

- The manuscript under review attempts to assess Women’s Perception of Male Involvement in Antenatal, Childbirth and Postnatal Care in Urban Slum Areas in Bangladesh. The study is of sound design and of clear practical and clinical interest.

- I found the article interesting and I think it could be of interest to Healthcare readership. 

- A brief introduction with the explanation of the study design is needed.

- Results are clearly presented and discussion and conclusion focus on the relevant topic and are supported by the results

- English grammar and style are fine. It is written without logical or factual errors.

- Please consider some articles to widen your introduction about the nutritional habits (DOI: 10.3390/nu12123688)

- Limitations and strength of the study need to be added before conclusion as a subsection of discussion.

Author Response

Response to the comments of reviewer #2:

  1. Please consider some articles to widen your introduction about the nutritional habits (DOI: 10.3390/nu12123688)

Response: We appreciate the suggestion of the reviewer, but unfortunately we didn’t find relevant literature suitable for our manuscript regarding women’s Perception of Male Involvement in reproductive health. However, we hope to address the issue of nutritional habits in our future study.

Reviewer 3 Report

Dear authors,

Thank you very much for creating this work and making this interesting subject discussable.

I only have 2 minor comments in order to improve the quality of this paper.

Wish you the best in publishing process

Best regards

Reviewer

Abstract and also conclusion sections

The authors used the abbreviation for “information, education, and communication (IEC)”. However this terminology was used only once and thus it may be omitted.

Please make sure that the abbreviations that are used in the tables, are also fully described. By doing this, both the text and the tables would become more comprehensive and understandable.

Author Response

Response to the comments of reviewer #3:

  1. Abstract and also conclusion sections: The authors used the abbreviation for “information, education, and communication (IEC)”. However, this terminology was used only once and thus it may be omitted.

Response: Thank the reviewer’s helpful suggestions. We have omitted this terminology from the abstract and conclusion sections.

  1. Please make sure that the abbreviations that are used in the tables are also fully described. By doing this, both the text and the tables would become more comprehensive and understandable.

Response: We thank the reviewer for kindling pointing out this issue. We mentioned the full form of the abbreviations used in the tables as the note under the tables.

Round 2

Reviewer 2 Report

I am satisfied with the revised version of the manuscript

This manuscript is a resubmission of an earlier submission. The following is a list of the peer review reports and author responses from that submission.